# Impacts on Crash Cushions—Analysis of the Safety Performance of Passenger Cars with Improved Safety Equipment Compared with Test Vehicles Based on Assessment Criteria as Defined in EN 1317

**Ernst Tomasch** * and **Gregor Gstrein**

Vehicle Safety Institute, Graz University of Technology, Inffeldgasse 13/VI, 8010 Graz, Austria;
gregor.gstrein@tugraz.at
\* Correspondence: ernst.tomasch@tugraz.at

**Abstract:** To assess the safety performance of crash cushions, guidelines or standards are used. Real-life accident conditions might deviate substantially from the approval test conditions. The objective of this study is to evaluate occupant safety in passenger cars in the event of an impact against a crash cushion. Real-life accident configurations deviate significantly from the impact configurations used in the approval test EN 1317. In four different tests, two vehicles regularly used in EN 1317 and two vehicles with improved safety equipment (airbag, pretensioner, and load limiter) are used. The impact speed is 100 km/h, whereas the crash cushion is designed for an impact speed of 80 km/h. One configuration is defined as a full overlap, and one has a 50% offset. The ASI (Acceleration Severity Index), THIV/OIV (Theoretical Head Impact Velocity/Occupant Impact Velocity), and PHD/ORA (Post Head Deceleration/Occupant Ride down Acceleration) are calculated from the acceleration signals. The offset impact was more serious for both the regularly used vehicle and the vehicle with improved safety equipment. Vehicles with improved safety equipment do not have any influence on these criteria. It is apparent that new occupant safety technologies will not have any influence on occupant safety performance. The criteria currently in use are more likely to be of use for assessing vehicle performance rather than occupant safety.

**Keywords:** crash cushion; occupant safety; EN 1317; ASI; THIV; PHD; OIV; ORA

## 1. Introduction

On high priority roads such as motorways, crash cushions are typically used to prevent errant vehicles from impacting hazardous obstacles on the side of the road (e.g., tunnel lay-bys, tunnel portals, and gore points; Figure 1).

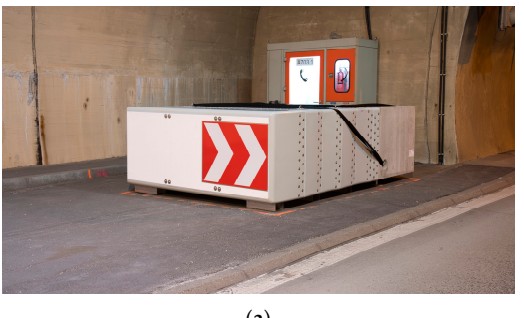
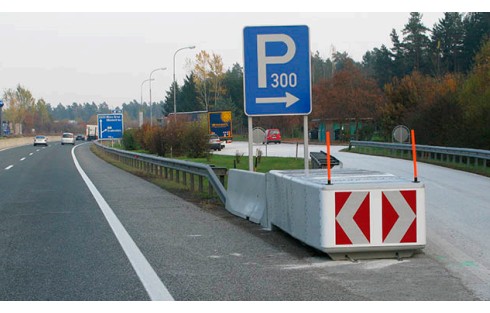

| (**a**) | (**b**) |

**Figure 1.** A crash cushion positioned in front of the end-wall of a lay-by (**a**) and a gore point (**b**) [1].

In order for crash cushions to be allowed to be installed in Europe, they must meet the requirements defined in the norm EN 1317 [2,3]. Within EN 1317, specific test specifications

are described. Depending on the design of the crash cushion (redirective or non-redirective), up to six tests with different vehicle classes, impact configurations (angle and overlap), and impact speeds are required to obtain European approval for a specific performance class. According to the EN 1317 test conditions, the vehicles that are used in testing must be recently produced models and, in the case of passenger cars up to and including a weight of 1500 kg, models commonly used in current traffic in Europe. The minimum test weight of the vehicles is 900 kg. No modifications (e.g., reinforcements) are permitted that would alter the general characteristics of the vehicle or invalidate its certification. However, according to the criteria of EN 1317 [2,3], vehicles used for the (certification) testing of crash cushions do not necessarily have to be state-of-the-art. Usually, the vehicles used are very old, and the airbags are deactivated during the test. For example, the average age of the vehicle fleet in Germany was 9.3 years in 2017 and is predicted to increase to 10 years in 2023 [4]. Nevertheless, the vehicles in an impact test are, in comparison, still significantly older, as an analysis of in-house test data from Graz University of Technology illustrates (Figure 2). The average age of the passenger cars in impact tests with vehicles of a mass of 900 kg is 21.6 (SD = 4.0) years, with a total range of 12.1 to 31.8 years. Data on the average age of the vehicles used in other test laboratories are not available.

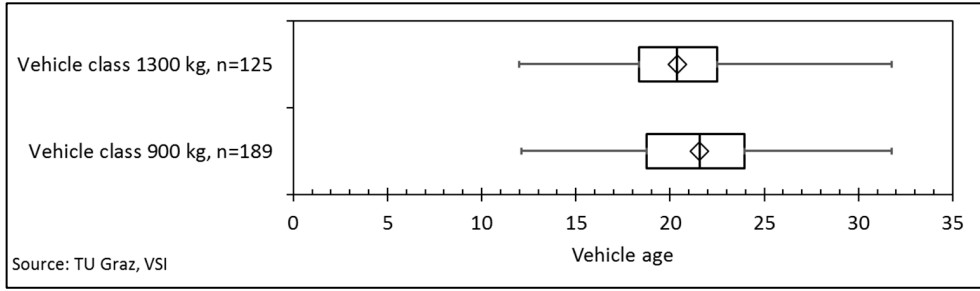

**Figure 2.** Average vehicle age of passenger cars used for impact tests according to EN 1317.

The International Council on Clean Transportation Europe [5] showed in their report that the mass of new registered vehicles in the EU has increased over the past 20 years, with even the smallest of these vehicles being above 1000 kg. Hernández et al. [6] investigated the number of new registered passenger cars in Spain with weights lower than 1000 kg. Only ten percent fall within this category. No information is given on how many vehicles with a mass of 900 kg are registered.

An investigation of the literature shows that occupants in older cars have a higher risk of sustaining severe or fatal injuries [7–11]. Kahane [12] analyzed US road accident data and compared the reported number of fatalities to the potential number of fatalities without vehicle safety technologies, i.e., the number of lives saved by safety technologies. Newer cars save more lives than older cars, and the risk of being fatally injured decreases in a newer car. The most important safety technology is the seat belt (including 3-point, lap-only, and automatic seat belts), with these having saved more than half of the lives. Pretensioners and load limiters for seat belts were not analyzed separately. Different studies (e.g., [13–16]), however, suggest they have a positive effect on the occupants, i.e., a reduction in the risk of injury. Another important safety technology is the air bag (frontal and side air bags). This technology is the second most important device for saving lives [12] and reducing the risk of injury [8].

In addition to occupant restraint systems and other influencing factors, a vehicle's weight has a significant influence on the risk of injury [17–19], and both gender and age [20,21] are also significant factors.

It can be assumed that vehicles with improved safety equipment (airbags, belt pretensioners, belt force limiters, etc.) and higher vehicle masses provide a lower risk of injury in the event of a collision as compared to vehicles that meet the criteria of EN 1317 [2] but have either deactivated or not installed adequate improved safety equipment.



The objective of this study is to assess occupant safety in passenger cars in the event of a collision with a crash cushion in which the impact configurations deviate from those of the approval tests. The assessment criteria for occupant safety are chosen according to EN 1317 [2] and the Manual for Assessing Safety Hardware (MASH) [22]. Crash cushions that have already been tested and certified in accordance with EN 1317 are investigated. In the tests, vehicles with improved safety equipment (airbags, seat belt pretensioners, and load limiters) are considered and compared to vehicles that are regularly used in EN 1317 tests. In real-life accidents, the impact configuration does not necessarily meet the configurations defined in EN 1317. Thus, the impact scenarios differ significantly from the impact configurations according to EN 1317 and are therefore not comparable with a certification test.

## 2. Materials and Methods

### 2.1. Experimental Set-Up

In the tests, a non-redirective F1-80 crash cushion from the manufacturer ALPINA (Bavaria, Germany), which was not attached to the road surface, was used. The performance level of this crash cushion is 80. The maximum test speed in the approval test is 80 km/h. The length of the crash cushion is 3600 mm, and the width is 2380 mm (Figure 3). The back-up is separated into two concrete blocks. The weight of the cushion bag is approximately 300 kg, and the back-up has a weight of approximately 6640 kg.

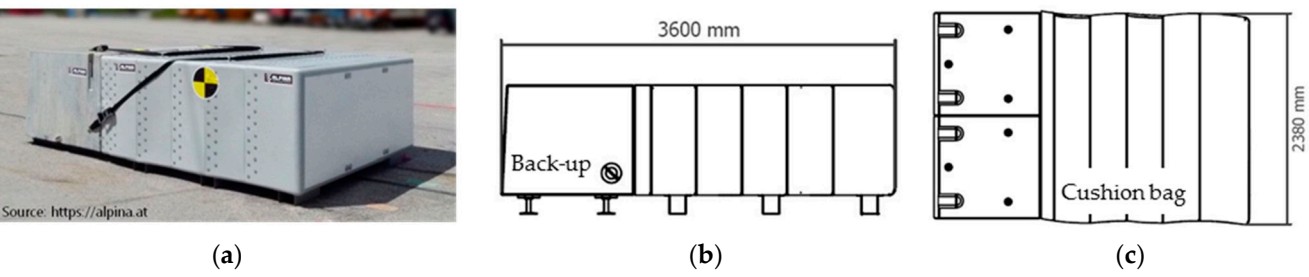

| (**a**) | (**b**) | (**c**) |

**Figure 3.** (**a**) A non-redirective crash cushion unmounted to the road surface, (**b**) side view, (**c**) top view [23].

The crash cushion was positioned according to the F1-80 installation manual (Figure 4). The gap between the crash cushion and the concrete wall was approximately one meter and was chosen in accordance with the tests performed by Tomasch et al. [24]. The angle of installation to the roadway was 5°. The run-off-road angle was calculated from the CEDATU (Central Database for In-Depth Accident Study) road accident database [25,26] and was set to 5° [24]. In combination with the installation angle, the impact angle calculates to zero degrees.

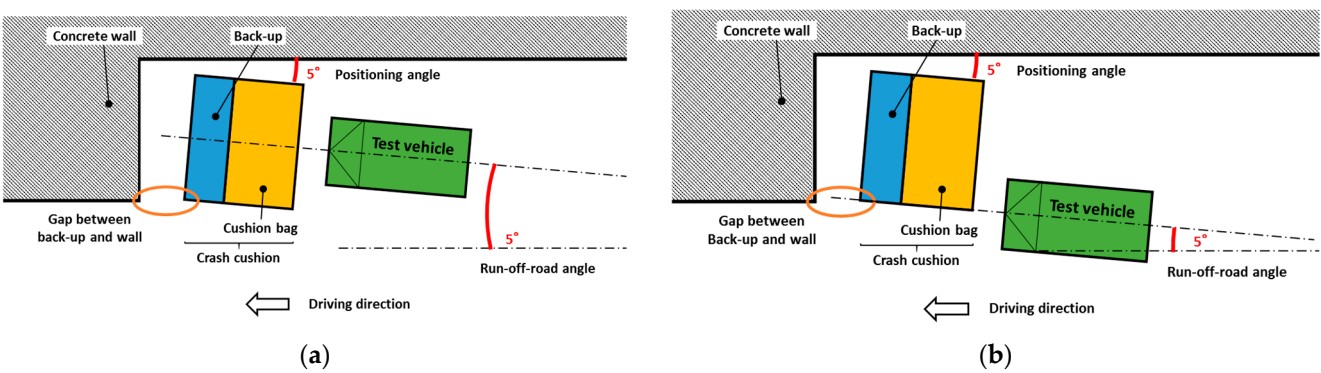

| (**a**) | (**b**) |

**Figure 4.** Test set-up: (**a**) vehicle position at the center line of the crash cushion at first contact and (**b**) vehicle offset with the vehicle's center line impacting the corner of the crash cushion.

The vehicle position at the point of impact was defined at the center line of the crash cushion, i.e., the center line of the vehicle and the crash cushion were coincident. The vehicle had a full overlap of 100%. A vehicle offset of 50% was defined for the second impact configuration. The center line of the vehicle impacts the corner of the crash cushion.

The impact speed was set to 100 km/h based on a real-life accident analysis of the CEDATU road accident database in accordance with Tomasch et al. [24]. The speed was measured approximately 2 m before the vehicle collided with the crash cushion.

For the tests with the vehicles regularly used in EN 1317 [3], two Opel Corsas with a total weight of 935 kg and 937 kg, respectively (incl. an anthropometric test device 75 kg), were used. The first registration date of the vehicles was month 10 of 1997 and month 10 of 1998, respectively. The vehicles were equipped with airbags and seat belt pretensioners; however, the airbag was deactivated for the test. The seat belt pretensioner was not deactivated.

For the tests with improved safety equipment, two VW Golf 6 vehicles with a total weight of 1245 kg each (incl. anthropometric test device 75 kg) were used. The year of manufacture of the vehicles was 2009, in month 3 and month 4, respectively. Both vehicles were equipped with frontal airbags, knee airbags, seat belt pretensioners, and belt force limiters. The airbags were active during the test.

A hybrid III 50th percentile male dummy (anthropometric test device (ATD)) with a mass of 75 kg was placed in the driver's seat. The ATD represents a vehicle occupant's size, shape, and mass.

Four tests were performed. The test matrix is given in Table 1. Unfortunately, in test 3, the emergency brake system was activated for some reason just before impact. The vehicle began a full deceleration during the final meters before colliding with the crash cushion. The measured speed two meters ahead of the crash cushion was 99.45 km/h and, in conjunction with the road conditions, the collision speed was calculated at 98 km/h.

**Table 1.** Test matrix of the performed tests (* impact speed determined to be 98 km/h).

| | Unit | Test 1 | Test 2 | Test 3 | Test 4 |
|---|---|---|---|---|---|
| Specimen | - | ALPINA F1-80 | ALPINA F1-80 | ALPINA F1-80 | ALPINA F1-80 |
| Vehicle | - | VW Golf 6 | Opel Corsa | VW Golf 6 | Opel Corsa |
| Year of manufacture | - | 2009.03 | 1997.10 | 2009.04 | 1998.10 |
| Weight incl. ATD | [kg] | 1245 | 935 | 1245 | 937 |
| Length | [m] | 4.21 | 3.74 | 4.21 | 3.74 |
| Width | [m] | 1.79 | 1.62 | 1.79 | 1.62 |
| Wheelbase | [m] | 2.574 | 2.443 | 2.574 | 2.443 |
| Airbag | - | activated | activated | activated | activated |
| Pretensioner | - | yes | yes | yes | yes |
| Load limiter | - | yes | no | yes | no |
| Target speed | [km/h] | 100 | 100 | 100 | 100 |
| Measured collision speed | [km/h] | 101.12 | 101.04 | 99.45 * | 100.28 |
| Gap of specimen to the wall | [cm] | 100 | 102 | 100 | 99 |
| Impact angle | [°] | 0.1 | 0.9 | 0.1 | 0.5 |
| Overlap | [%] | 100 | 100 | 50 | 50 |
| Road conditions | - | dry | dry | wet | slightly wet |

*2.2. Data Acquisition*

Two acceleration sensors (manufacturer: ASC, Pfaffenhofen an der Ilm, Germany; type: 5411LN-100 and Measurement Specialties; type: 1203-0500-10-240X) and one angular velocity sensor (manufacturer: IES, Kirchheim bei München, Germany; model: 2103-2400) were used to record the acceleration and the angular velocity of the vehicle body. The measuring range of the acceleration sensors is 100 g and 500 g, respectively. The measurement range of the angular velocity sensor is 2400°/s. All sensors were mounted on a metal plate, which was attached to the vehicle (Figure 5) as near to its center of gravity

as possible. All sensors were calibrated. The sensors' positions and the coordinate system of the vehicle were defined according to EN 1317 [3].

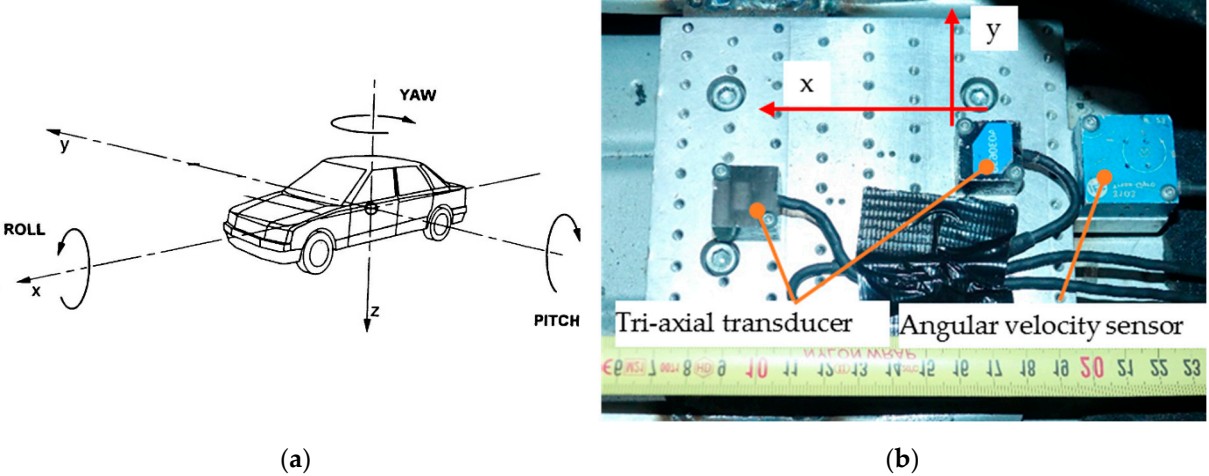

(**a**)                                               (**b**)

**Figure 5.** Vehicle coordinate system (**a**) and position of the acceleration transducers mounted at the center of gravity (**b**) c.f. [24].

A MiniDAU (Mini Data Acquisition Unit) from Kayser Threde (Munich, Germany, K3700 MiniDAU®) was used for measurement data acquisition. The recording rate was 10 kHz. The measurement data were synchronized via a contact switch in the front of the vehicle.

The tests were recorded with three high-speed cameras at 500 frames per second. The positions of the cameras are shown in Figure 6. One camera was positioned laterally (1). A second camera recorded the test from an oblique angle (2). An overhead camera (3) recorded the vehicle's motion, and the fourth camera (frame rate: 24 frames per second) also recorded the vehicle's approach prior to impact (4).

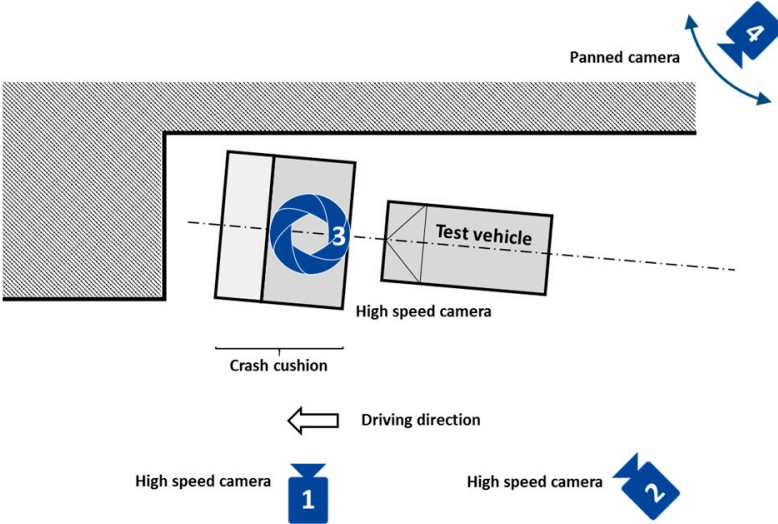

**Figure 6.** Sketch of the position of the video cameras. Positions 1 to 3: high speed cameras. Position 4: panned normal speed camera.

*2.3. Assessment*

Occupant safety was assessed by evaluating criteria derived from the acceleration and rotational speed of the vehicle body:

- ASI (Acceleration Severity Index);
- THIV (Theoretical Head Impact Velocity);

- PHD (Post Head Deceleration);
- OIV (Occupant Impact Velocity);
- ORA (Occupant Ride down Acceleration);
- Delta-v (instantaneous change of velocity).

The ASI and THIV are assessed according to the European Standard EN 1317 [2], and the OIV and ORA are computed to assess the occupant injury risk in MASH [22]. Though not required, the ASI, THIV, and PHD indices shall be calculated for safety hardware sold in the US [22]. However, in the latest edition of EN 1317 [27], the PHD is no longer required.

Although the European Committee for Standardization (CEN) does not provide background details on the ASI, the calculation of the "Severity Index" (SI) by Weaver and Marquis [28,29] looks similar to that of the ASI. The ASI is calculated from the three components (x, y, and z) of the acceleration sensor at the vehicle's center of gravity [3]. Since only the vehicle's acceleration is used in the calculation, the ASI assumes that the occupant is in direct contact with the vehicle for the entire duration of the acceleration phase.

$$\text{ASI(k)} = \left[ \left( \frac{\overline{A_x}}{12} \right)^2 + \left( \frac{\overline{A_y}}{9} \right)^2 + \left( \frac{\overline{A_z}}{10} \right)^2 \right]^{0.5}. \tag{1}$$

The THIV, OIV, PHD, and ORA are based on the flail space model developed by Michie [30]. The flail space model assumes that occupant injury severity depends on the impact velocity at which the occupant hits parts of the passenger compartment and the subsequent deceleration acting on the occupant. An unbelted occupant is assumed to travel a distance of 0.6 m longitudinally and 0.3 m laterally in free flight ("flail") before impacting an interior part of the passenger compartment. For the OIV, vehicle rotation and pitching motion are not taken into account. The OIV corresponds to a theoretical speed of the head when impacting the vehicle interior. The actual value is calculated at the moment when the unbelted occupant passes either the longitudinal or lateral boundary. After the impact, the occupant is assumed to remain in contact with the vehicle parts and decelerate with the vehicle. The resulting acceleration is used to calculate the ORA.

A simplified geometry of the vehicle's interior is also assumed for determining the THIV. Contrary to the OIV, however, the THIV takes into account the yaw angle velocity. Subsequently, the following formula is used to calculate the THIV, wherein the x and y indices represent the velocity components at the vehicle's center of gravity [3].

$$\text{THIV} = \left[ V_x^2(T) + V_y^2(T) \right]^{0.5}. \tag{2}$$

To calculate PHD or ORA, it is assumed that the head of the anthropomorphic test device collides with the vehicle's interior during the collision and thus experiences the same acceleration as the vehicle for the rest of the contact phase. The PHD value represents the maximum resulting acceleration of the components in the x and y-directions, measured at the center of gravity. The PHD, however, is no longer used in EN 1317 [3].

$$\text{PHD} = \max \left[ (a_x(t))^2 + (a_y(t))^2 \right]^{0.5}. \tag{3}$$

The change of velocity during a collision (delta-v) is a metric that represents the severity of injury of a vehicle's occupants. It describes the change in velocity that occurs as a result of the impact. Delta-v is a vectorial quantity and thus has a magnitude and direction, and it is calculated using the velocity at the moment of the impact and the release velocity after the impact. In ISO 12353-1 [31], the velocity change due to collision is defined as the "*vector difference between impact velocity and separation velocity*".

### 3. Results

*3.1. Case by Case Study*

3.1.1. Test 1—VW Golf, Full Overlap Impact

The vehicle hit the center of the crash cushion with an overlap of 100%. The back-up of the crash cushion started to move after approx. 70 ms and hit the concrete wall with its rear edge after approx. 316 ms. The vehicle was fully decelerated and started to accelerate against the direction of impact at 450 ms. The final position was reached after 1.5 s. In its final position, the vehicle was rotated clockwise approximately 2° around its vertical axis.

During the impact, the vehicle moved beneath the crash cushion, and the crash cushion was lifted upwards. The bumpers were fully compressed, and the floor panel impacted the road. The vehicle hit the back-up of the crash cushion after approx. 150 ms, and the vehicle and the back-up continued to move at a constant speed against the concrete wall. At approx. 316 ms, the back-up collided with the concrete wall, and an increase in the acceleration signal was observed.

Approx. 20 ms after the impact, the front airbags (driver airbag, knee airbag, and passenger airbag) started to deploy (visible on the video). The side airbags (curtain and seat) did not deploy. The seat belt pretensioner and the seat belt load limiter were activated. Webbing marks on the seat belt were visible.

No intrusions into the passenger compartment were observed; the passenger compartment remained undamaged, and the wheelbase was not shortened. The damage occurred mainly in the area of the bonnet and was caused by the underrun of the vehicle. The impact on the back-up did not contribute significantly to the damage.

The ASI reached its maximum at approx. 110 ms with 1.60 (sensor 1) and 1.58 (sensor 2). A further peak was detected at approx. 434 ms (0.41 for sensor 1 and 0.4 for sensor 2), which was caused by the impact of the back-up against the wall. The THIV was calculated at 48.53 km/h and 48.35 km/h, respectively.

Pictures of the car's moment of impact, final position, and damage pattern are shown in Figure 7.

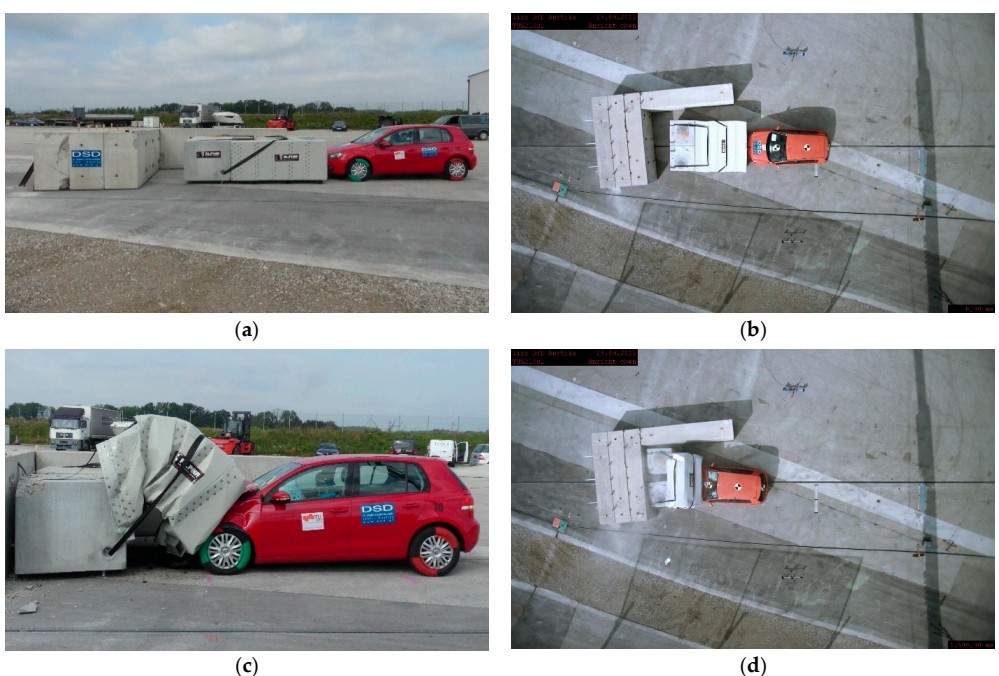

(**a**)          (**b**)

(**c**)          (**d**)

**Figure 7.** *Cont.*

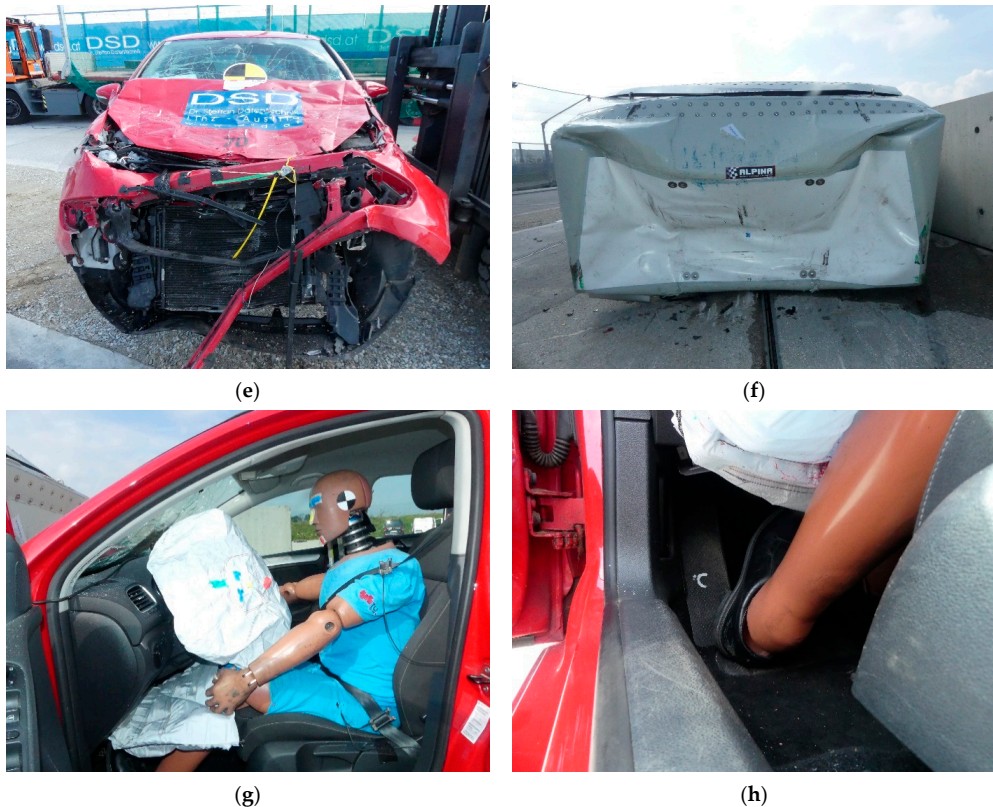

(e)                    (f)

(g)                    (h)

**Figure 7.** Test 1: Initial position of the vehicle (picture (**a**,**b**)). Final position of the vehicle after the impact (picture (**c**,**d**)). Damage to the vehicle and crash cushion (picture (**e**,**f**)). ATD position and lower legs (picture (**g**,**h**)).

### 3.1.2. Test 2—Opel Corsa, Full Overlap Impact

The vehicle hit the center of the crash cushion with an overlap of 100%. The back-up of the crash cushion started to move after approx. 110 ms and hit the concrete wall with the rear edge after approx. 424 ms. The vehicle was fully decelerated and started to accelerate against the direction of impact at about 880 ms. The final position was reached after 1.1 s with a yaw angle of about 5° of clockwise rotation as compared to the impact position.

As in test 1, the vehicle was moved beneath the crash cushion during the impact, causing the crash cushion to be lifted slightly upwards. The bumpers of the vehicle were completely compressed, and the floor panel impacted the road surface. An impact of the vehicle against the back-up could not be determined, doubtlessly due to the dust generated during the test.

Approximately 98 ms after the impact, the head of the ATD hit the steering wheel. However, definite contact could not be identified in the video analysis.

There were no intrusions into the passenger compartment, but the header rails were buckled on the driver's side around the area of the B-pillar. As a result of the impact, the wheelbase was shortened by about 6 cm on both sides.

The seat belt pretensioner was activated, and webbing marks were visible on the seat belt. Knee contact between the ATD and the dashboard was observed.

The ASI reached the maximum at approx. 105 ms with 1.66 (sensor 1) and 1.64 (sensor 2). The THIV was calculated at 49.01 km/h and 48.95 km/h, respectively.

Pictures of the car's moment of impact, final position, and damage pattern are shown in Figure 8.

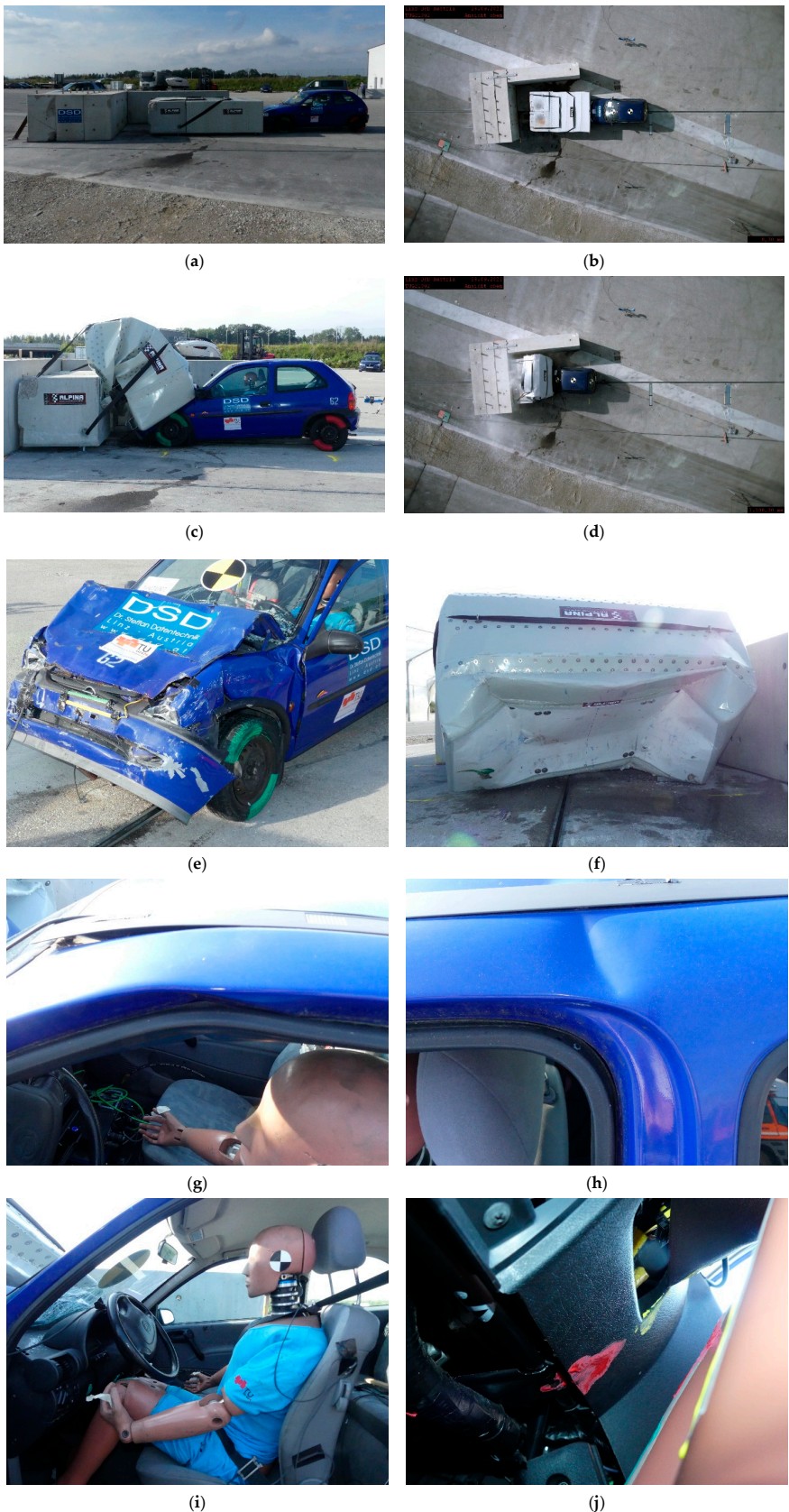

**Figure 8.** Test 2: Initial position of the vehicle (picture (**a**,**b**)). Final position of the vehicle after the impact (picture (**c**,**d**)). Damage to the vehicle and crash cushion (picture (**e–h**)). ATD position and lower leg contact against the dashboard (picture (**i**,**j**)).

### 3.1.3. Test 3—VW Golf, Offset Impact

The vehicle hit the crash cushion with an overlap of 50%. Half of the right front of the vehicle was covered. The back-up of the crash cushion started to move after approx. 100 ms, and hit the concrete wall with its rear edge after approx. 270 ms. The vehicle rotated clockwise around its vertical axis into its final position, coming to a complete standstill after 1.25 s against the direction of travel prior to the impact. The yaw angle in the final position was approx. 145° relative to the impact position. The distance between the left front wheel and the corner of the concrete wall was approx. 3.2 m, and the distance between the left rear wheel and the corner of the concrete wall was approx. 4.6 m.

The vehicle penetrated the cushion bag and crashed against the back-up after approx. 100 ms. The main damage is on the front right half of the vehicle, according to the impact configuration. The right front wheel contacted the wheelhouse. The right wheelbase was shortened by approx. 9 cm. On the left side, the wheelbase remained unchanged by the impact. No intrusions into the passenger compartment could be detected.

Roughly 128 ms after the impact, deployment of the front airbags (driver airbag, knee airbag, and front passenger airbag) became visible in the video. The side airbags (curtain and seat) did not deploy. The seat belt pretensioner and load limiter were activated. Webbing marks on the seat belt were visible.

The head of the ATD hit the steering wheel or was hit by the deploying airbag close to the steering wheel at approx. 128 ms.

The ASI reached the maximum at approx. 136 ms with 2.11 (sensor 1) and 2.09 (sensor 2). The THIV was calculated at 51.56 km/h and 50.70 km/h, respectively.

Pictures of the car's moment of impact, final position, and damage pattern are shown in Figure 9.

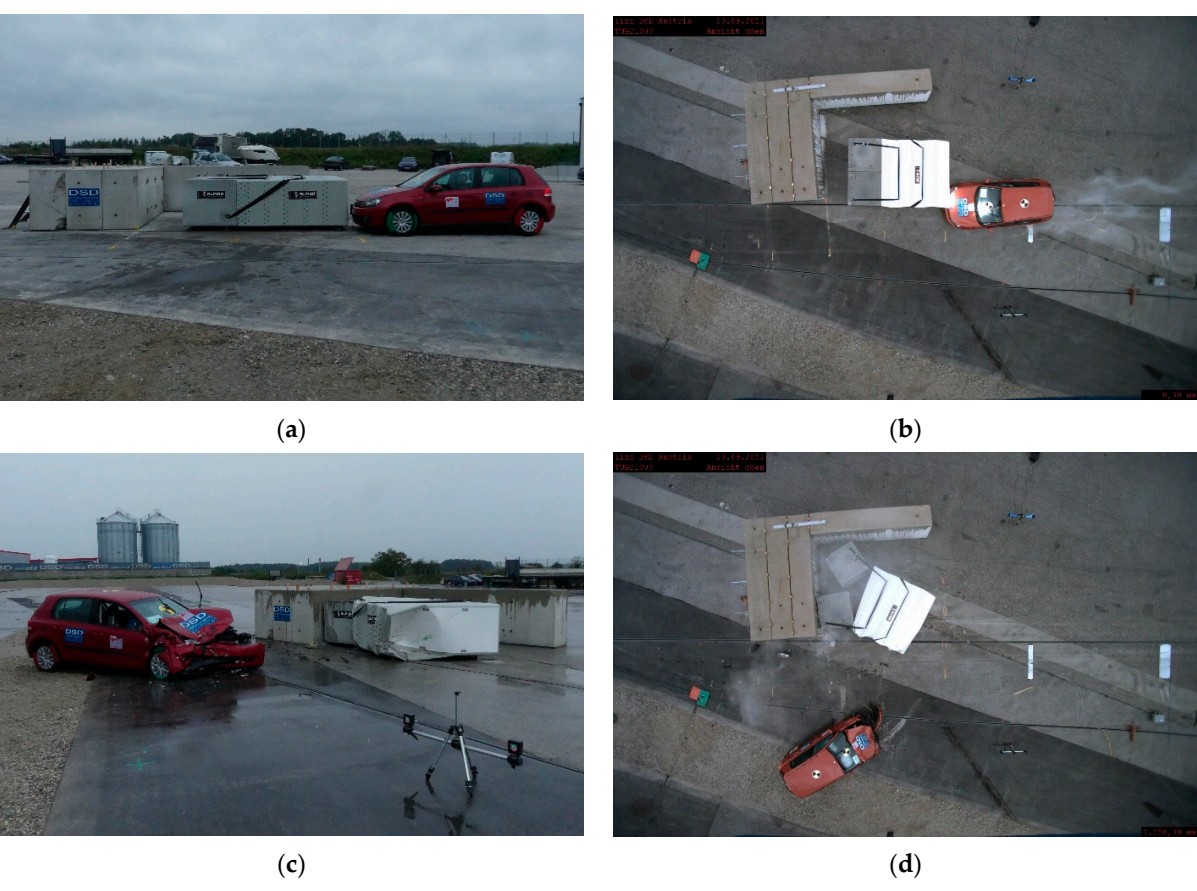

(**a**) (**b**)

(**c**) (**d**)

**Figure 9.** *Cont.*

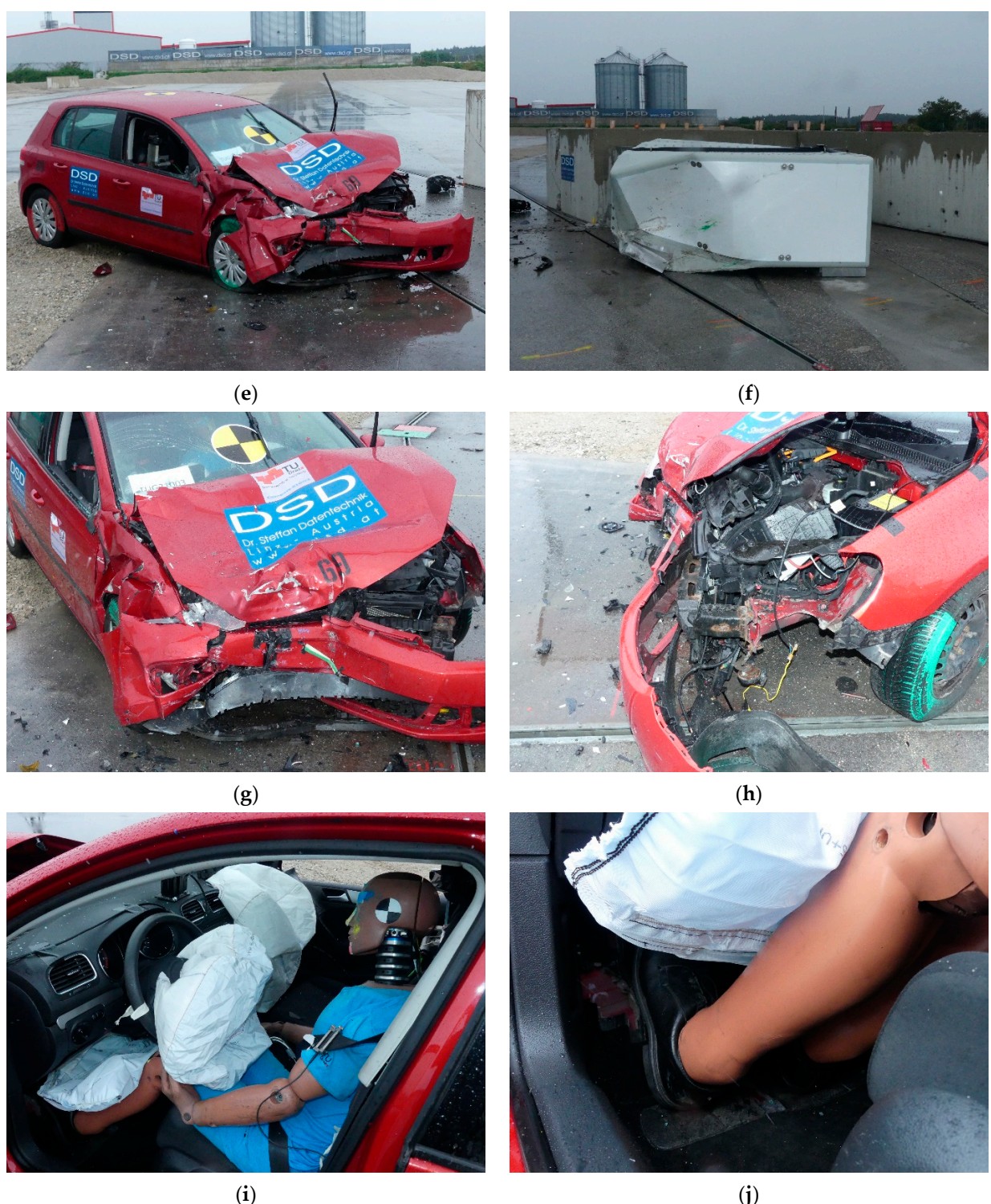

(**e**)

(**f**)

(**g**)

(**h**)

(**i**)

(**j**)

**Figure 9.** Test 3: Initial position of the vehicle (picture (**a**,**b**)). Final position of the vehicle after the impact (picture (**c**,**d**)). Damage to the vehicle and crash cushion (picture (**e**–**h**)). ATD position and lower leg position (picture (**i**,**j**)).

### 3.1.4. Test 4—Opel Corsa, Offset Impact

Similar to test 3, the vehicle hit the crash cushion with an overlap of 50% so that half of the right front of the vehicle was covered. The back-up of the crash cushion started to move after approx. 112 ms and hit the concrete wall with its rear edge after approx. 294 ms. The vehicle rotated clockwise around its vertical axis into the end position and came to a

complete standstill after 1.5 s against the direction of travel prior to the impact. The yaw angle in the end position was approx. 220° relative to the impact position. The distance between the left front wheel and the corner of the concrete wall was approx. 6.44 m and the distance between the left rear wheel and the corner of the concrete wall was approx. 5.66 m.

The vehicle penetrated the crash cushion and crashed into the back-up after approx. 108 ms. The damage was on the front right half of the vehicle, according to the impact configuration. The right front wheel contacted the wheelhouse. The right wheelbase was shortened by approx. 14 cm, and the left wheelbase was lengthened by approx. 6 cm. The right header rail at the B-pillar buckled slightly. Intrusions into the passenger compartment could not be detected.

The seat belt pretensioner was activated and webbing marks on the seat belt were visible.

The head of the ATD hit the steering wheel at approx. 143 ms. The impact could be clearly detected, with evidence (colored marks) appearing on the steering wheel.

The ASI reached the maximum at approx. 130 ms with 2.63 (sensor 1) and 2.60 (sensor 2). The THIV was calculated at 43.37 km/h and 43.29 km/h, respectively.

Pictures of the car's moment of impact, final position, and damage pattern are shown in Figure 10.

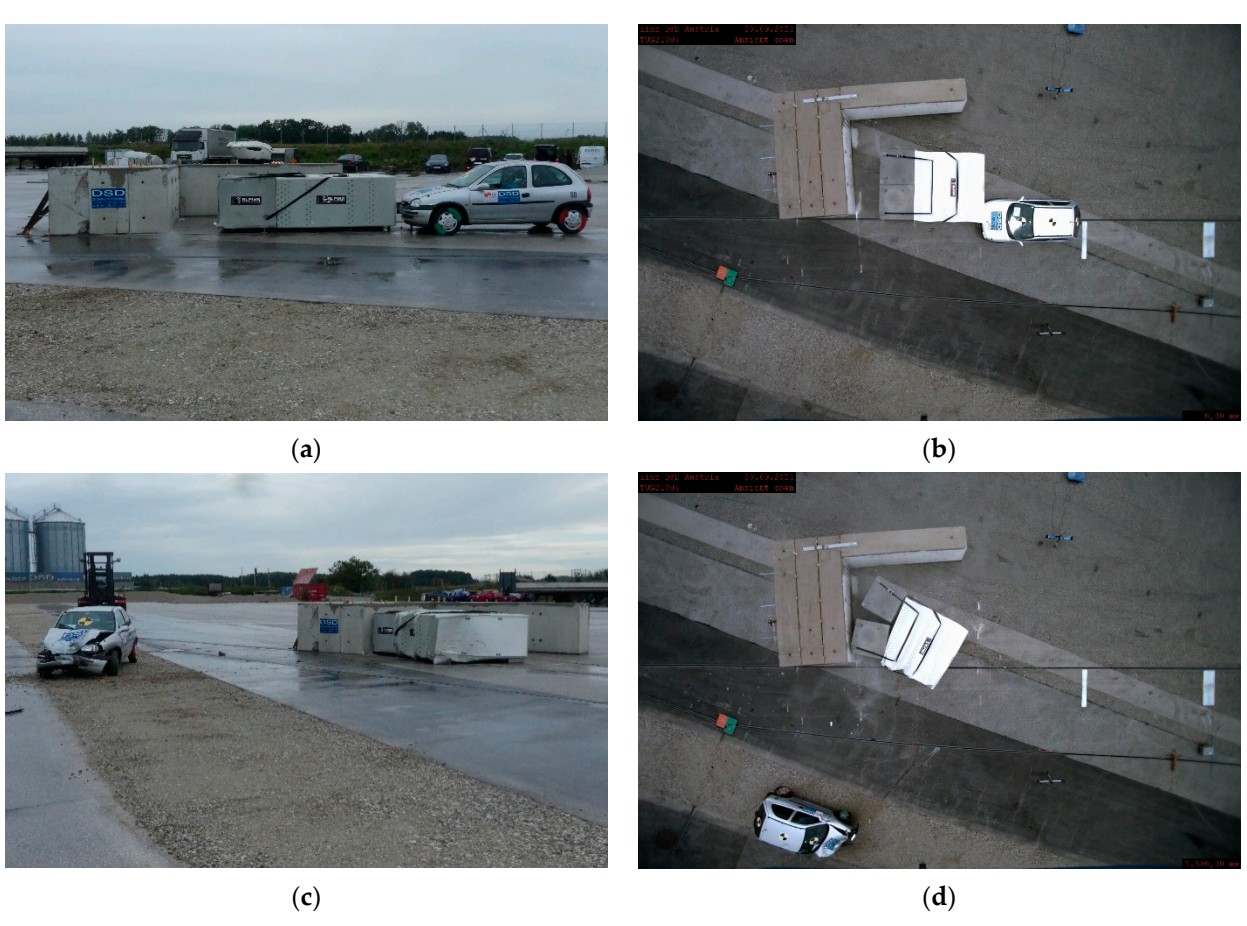

(**a**)          (**b**)

(**c**)          (**d**)

**Figure 10.** *Cont.*

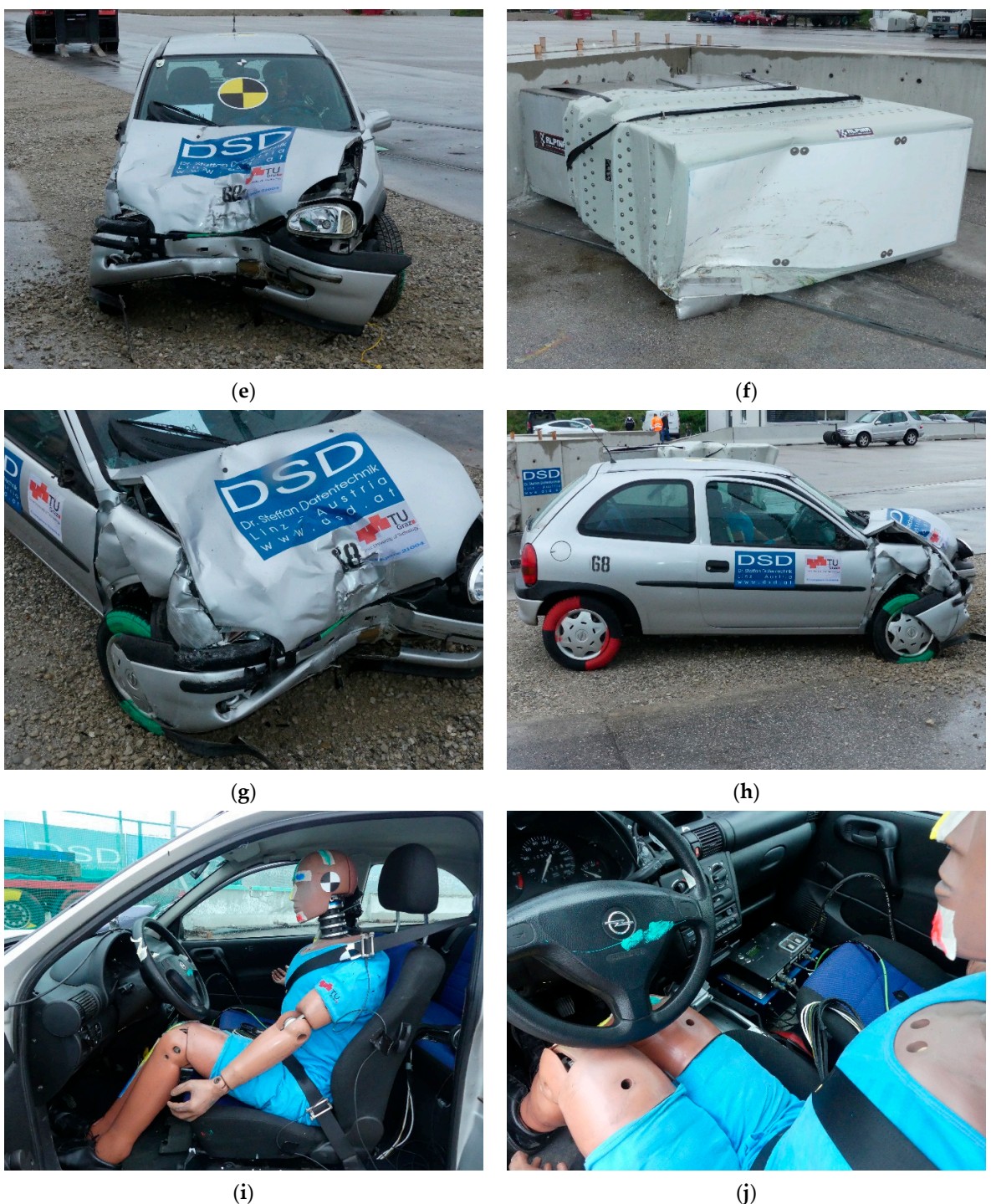

**Figure 10.** Test 4: Initial position of the vehicle (picture (**a**,**b**)). Final position of the vehicle after the impact (picture (**c**,**d**)). Damage to the vehicle and crash cushion (picture (**e**–**h**)). ATD position and head contact evidence on the steering wheel (picture (**i**,**j**)).

### 3.2. Injury Risk Assessment

In the case of an impact with a full overlap, there is a clear peak in the acceleration signal at the beginning of the impact, which cannot be detected in the offset impacts (Figure 11). After the peak, the acceleration decreases and then increases again. If the first acceleration peak is ignored, then the acceleration signal has a more or less trapezoidal characteristic. In the case of an impact with an offset impact, an acceleration of approx. 8 to 10 g is reached until the vehicle strikes the back-up, which is clearly visible in the

acceleration signal. From this point on (approx. 100 ms), the acceleration increases to approx. 30 to 40 g. The Opel Corsa experiences a significantly higher maximum acceleration (approx. 40 g) than the VW Golf (approx. 30 g). The acceleration characteristic is initially comparable to the full overlap, except at the beginning peak. After the impact with the back-up, a triangular shaped curve appears.

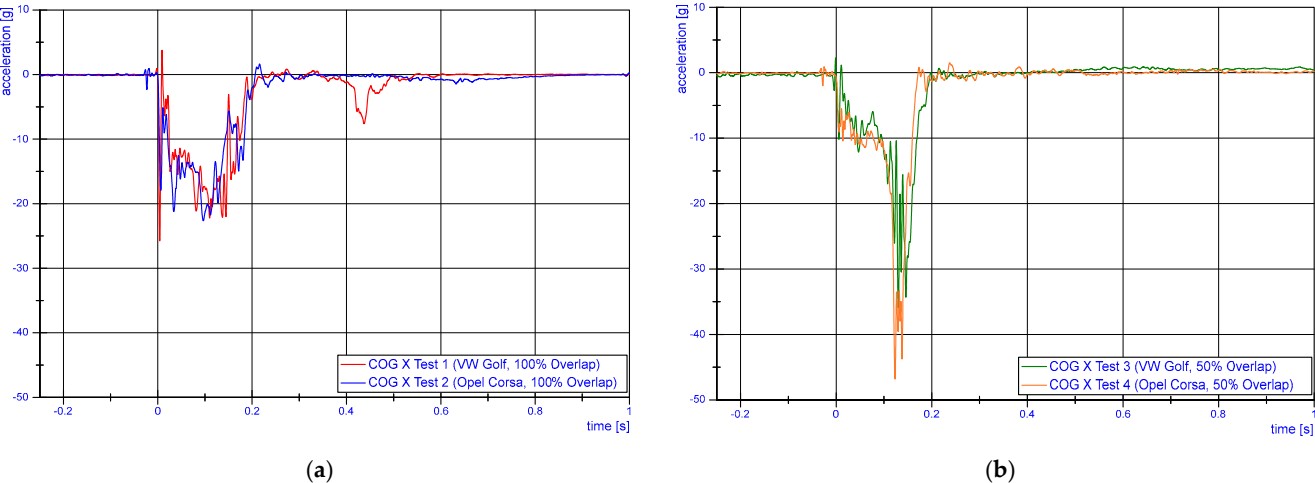

(**a**)          (**b**)

**Figure 11.** Vehicle acceleration time history in x-direction: (**a**) full overlap impact; (**b**) offset impact.

In both impact configurations (full overlap and offset impact), the acceleration time is approx. 200 ms, whereas in the case of the Opel Corsa in the offset impact, the acceleration time is approx. 170 ms.

In the offset impact, the vehicle was crushed against the back-up after approx. 100 ms (VW Golf) and approx. 108 ms (Opel Corsa). From this point on, the vehicle's acceleration started to increase significantly. At this point, a speed reduction of approx. 27 km/h and approx. 37 km/ is detected, respectively, for the VW Golf and Opel Corsa. Thus, the VW Golf collides with the back-up at a speed of approx. 73 km/h and the Opel Corsa collides with the back-up at a speed of approx. 63 km/h. In comparison, at the same time, a speed reduction of approx. 45 km/h and approx. 51 km/h is observed, respectively, for the VW Golf and the Opel Corsa during the full overlap tests.

At the end of the vehicle's acceleration (at approx. 200 ms), the delta-v in the longitudinal direction of the vehicle (y-component is not taken into account), is approx. 91 km/h for the VW Golf and approx. 93 km/h for the Opel Corsa in the full overlap test. In the case of the offset impact test, a change in collision speed of approx. 81 km/h and approx. 87 km/h could be determined, respectively, for the VW Golf and the Opel Corsa.

In Table 2, the resulting values for the different assessment criteria are summarized.

**Table 2.** Assessment criteria.

|  |  | Test 1 | Test 2 | Test 3 | Test 4 |
|---|---|---|---|---|---|
| Criterion | Unit | VW Golf Mk6 | Opel Corsa | VW Golf Mk6 | Opel Corsa |
| Delta-v | [km/h] | 91 | 93 | 81 | 87 |
| ASI (Sensor 1) | [-] | 1.60 | 1.66 | 2.11 | 2.63 |
| ASI (Sensor 2) | [-] | 1.58 | 1.64 | 2.09 | 2.60 |
| THIV (Sensor 1) | [km/h] | 48.53 | 49.09 | 51.56 | 43.37 |
| THIV (Sensor 2) | [km/h] | 48.35 | 48.95 | 50.70 | 43.29 |
| OIV (Sensor 1) | [km/h] | 48.52 | 49.09 | 48.13 | 43.90 |
| OIV (Sensor 2) | [km/h] | 48.37 | 49.03 | 47.53 | 43.74 |
| PHD (Sensor 1) | [g] | 20.94 | 20.76 | 30.83 | 42.41 |

**Table 2.** *Cont.*

|  |  | Test 1 | Test 2 | Test 3 | Test 4 |
|---|---|---|---|---|---|
| PHD (Sensor 2) | [g] | 20.81 | 20.67 | 30.80 | 42.22 |
| ORA (Sensor 1) | [g] | 20.94 | 21.78 | 31.90 | 42.41 |
| ORA (Sensor 2) | [g] | 20.81 | 21.67 | 31.78 | 42.22 |

The acceleration characteristic also has an effect on the ASI curve (Figure 12a). The slopes of the ASI curves in the offset impact tests are flatter as compared to the tests with full overlap. When the vehicles hit the back-up, there is a clear discontinuity in the curve. There is no significant difference between the ASI of the two vehicles in the full overlap tests. An ASI of 1.6 was calculated for the VW Golf and an ASI of 1.66 for the Opel Corsa. In the case of an offset impact, however, the ASI of 2.11 for the VW Golf is significantly lower than the ASI of 2.63 for the Opel Corsa.

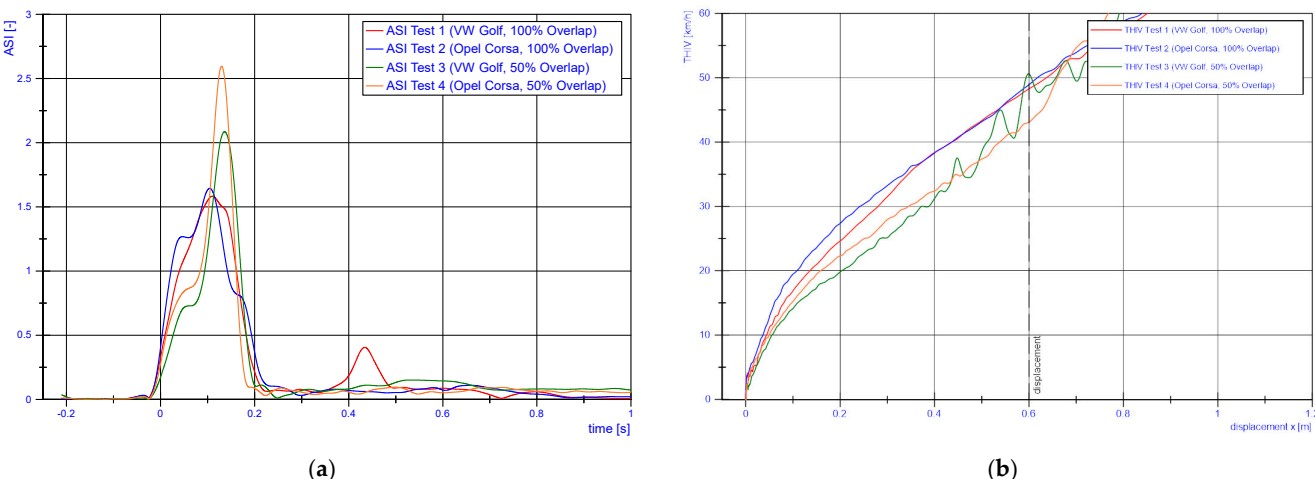

(**a**)                                                                    (**b**)

**Figure 12.** (**a**) ASI time history; (**b**) THIV displacement.

In any case, the ASI in the offset impact tests is significantly higher than in the full overlap tests, which results from the impact with the back-up after completely compressing the crash cushion.

The picture is somewhat different for the THIV (Figure 12b). In the tests with full overlap, the values do not differ significantly. For the VW Golf, a THIV of 48.53 km/h was recorded and, for the Opel Corsa, a THIV of 49.09 km/h was recorded. In the case of an offset impact, THIVs of 51.56 km/h and 43.37 km/h were recorded, respectively, for the VW Golf and the Opel Corsa. Similar results can be observed for the OIV.

The PHD is 20.94 g for the VW Golf with full overlap and 20.76 g for the Opel Corsa, and thus does not differ significantly between the two. For an offset impact, the PHD is significantly lower for the VW Golf at 30.83 g as compared to the Opel Corsa at 42.41 g. The same tendency was found for the ORA. At full overlap, the ORA is 20.94 g for the VW Golf and 21.78 g for the Opel Corsa. In the case of an offset impact, the ORA for the VW Golf (31.9 g) is significantly lower than for the Opel Corsa (42.41 g).

PHD and ORA for the full overlap are significantly lower when compared to the offset impact of both vehicles.

## 4. Discussion

The analyzed acceleration signals differ significantly between the full overlap impact and the offset impact. In the case of an impact with full overlap, immediately after the impact, a steep increase in the acceleration signal is detected and a local maximum of approx. 27 g (VW Golf) and approx. 18 g (Opel Corsa) is reached. The acceleration then

drops to 5 g and increases continuously to a maximum acceleration of approx. 22 g for both the VW Golf and the Opel Corsa. In the tests with an offset impact, an acceleration of approx. 10–12 g is observed for both vehicles in the first 100 ms. After this initial period, the vehicle hit the back-up, causing the acceleration to increase to the maximum values (VW Golf approx. 38 g and Opel Corsa approx. 48 g). In the full overlap test, the airbag began to deploy after approximately 20 ms, but in the offset impact, the airbag began to deploy after approx. 128 ms. Compared to the full overlap test, the acceleration level of the first time period is relatively low, and there is no significant gradient at the beginning that would initiate the deployment of the airbag. Thus, the airbags do not provide any safety protection for the occupant during the offset impact, as the airbag deployed directly before the head hit the steering wheel. This effect could actually be disadvantageous to the occupants.

In the offset impact, a speed reduction of approx. 27 km/h and approx. 37 km/h is observed, respectively, for the VW Golf and the Opel Corsa within the first 100 ms. This is the time stamp at which the vehicles hit the back-up of the crash cushion. Because of the cushion, the speed of the vehicle is reduced to 73 km/h for the VW Golf and 63 km/h for the Opel Corsa. It can be easily observed that the VW Golf's greater mass leads to a lower speed reduction before impacting the back-up.

For the ASI criterion, no significant difference can be seen between the two vehicles in the case of full overlap. The ASI is 1.6 for the VW Golf and 1.66 for the Opel Corsa. Due to the collision of the vehicles with the back-up with an offset impact configuration, the ASI for both vehicles is higher as compared to the full overlap configuration (VW Golf: 2.11 and Opel Corsa: 2.63). In both cases, however, the ASI for the lighter vehicle is higher than for the heavier one. Specifically, the difference is much higher for the offset impact. Following the study by Gabauer and Gabler [32] on injury risk curves related to ASI, an ASI of 1.6 (VW Golf, full overlap) would be expected to result in an MAIS 2+ injury with a risk of 48%. The risk of an MAIS 3+ injury would be 15%. For the Opel Corsa with full overlap with an ASI of 1.66, the risk of an MAIS 2+ injury would be 50.5% and an MAIS 3+ injury 16.5%. In the case of an offset impact, the VW Golf would have a risk of 70.5% for MAIS 2+ injuries and 29.0% for MAIS 3+ injuries. For the Opel Corsa, a risk of 88.0% for MAIS 2+ and 50.5% for MAIS 3+ injuries would be expected. Gabauer and Gabler [32] note, however, that the traffic accident data studied for this purpose came from only one vehicle manufacturer. Nevertheless, they do not expect a large variance when considering other manufacturers in their results.

In the case of the THIV, there is also no difference between the two vehicles for the full overlap. The THIV for the VW Golf is 48.5 km/h, only slightly below the THIV of the Opel Corsa at 49.1 km/h. Both signals show a very similar characteristic. The THIV for the offset impact of the VW Golf is significantly higher (51.6 km/h) as compared to the Opel Corsa (43 km/h). After a distance of approx. 42 cm, "artifacts" can be observed in the signal which are not present in the other tests. The THIV is calculated at a distance of 0.6 m. If the signal had the same characteristics as in the other tests, the THIV would fall in the range of 45 km/h to 47 km/h. This means the THIV would have the tendency to be lower in the offset impact than in the full overlap configuration. According to Gabauer and Gabler [32], the risk of MAIS 2+ injuries would be approx. 60% and MAIS 3+ injuries approx. 20% in a test with full overlap and a THIV of approx. 49 km/h. The risk of MAIS 2+ injuries for an offset impact in the VW Golf would be 66% and MAIS 3+ injuries 25%. For the Opel Corsa (THIV approx. 44 km/h), the risk of MAIS 2+ injuries would be approx. 48% and the risk of MAIS 3+ injuries approx. 14%.

The collision speed is almost completely dissipated during impact with the crash cushion, specifically in the test with the full overlap. The delta-v is between 81 km/h and 93 km/h, depending on the impact configuration. Based on risk functions for frontal collisions according to Augenstein et al. [33], MAIS 3+ injuries with a risk of 90% and above would be expected. In real-life accident configurations against crash cushions, the occupants often sustain no injuries. Although the delta-v is a very good predictor for

the assessment of injury risks, it seems delta-v is not a suitable parameter for assessing crash cushions.

A clear correlation between the vehicle's mass and the assessed criteria cannot be derived from the present tests. In the full overlap configuration, none of the assessed criteria differed significantly between the two vehicles. The influence of the vehicle's mass might be compensated for by the different stiffnesses of the vehicles. The Opel Corsa (old vehicle) absorbed more deformation (wheelbase shortened and header rail buckled) as compared to the VW Golf, which did not have a shortened wheelbase or other area that buckled; thus, the Opel Corsa is somewhat "softer" as compared to the VW Golf. If a vehicle with a similar mass to the VW Golf were used instead of the Opel Corsa, lower values would be expected for the criteria analyzed for older vehicles. In the offset impact, however, there were considerable differences in the criteria. The ASI, PHD, and ORA were found to be lower for the heavier vehicle, whereas the THIV and OIV were found to be higher. Probably because of the complex interaction between the stiffness and weight of the vehicle, no further conclusion with respect to the criteria could be made. However, in a study by Burbridge et al. [34], a tendency toward lower ASI, OIV, and ORA values was found in accordance with increases to mass. The vehicle mass therefore has a positive effect on the vehicle criteria. This could not be clearly observed in the tests carried out.

## 5. Limitations

Although the tests provided some promising results, relatively few tests were performed. In fact, only one vehicle was tested in each test configuration. With increases to mass, lower values for the analyzed criteria would be expected [34]. However, this could not be clearly concluded from the present tests. Thus, the test matrix would need to be extended in order to be able to evaluate the influence of vehicle mass on the analyzed criteria in more detail.

## 6. Conclusions

The certification of vehicle restraint systems (crash cushions) was carried out according to precisely define impact configurations in accordance with EN 1317 [2]. These standardized tests, however, can deviate significantly from the accident situation. The vehicles used in the certification tests are usually very old and do not correspond to the latest state-of-the-art. Therefore, vehicles with improved safety equipment (airbags, belt force limiters, etc.) were compared with vehicles that are used regularly in the certification tests, applying the assessment method as defined in EN 1317 [3].

In the offset impact configuration, both vehicles—the regularly used vehicle in the EN 1317 tests and the vehicle with improved safety equipment—had significantly inferior performances as compared to the full overlap configuration. The current offset impact test in the EN 1317 is defined with a 25% offset of the vehicle. If the crash cushion has enough width, the vehicle will still collide with a full overlap in the test. To improve road safety, it is suggested that the specific test be modified in such a way that the crash cushion is hit by the vehicle with an offset of, e.g., 50%.

Despite the vehicle year of manufacture differing by almost twelve years in the full overlap, the analyzed criteria showed no significant differences. In the offset impact configuration, however, the more recent vehicle performed better for the ASI and PHD (or ORA) but performed worse for THIV or OIV.

The literature study showed that newer vehicles and vehicles with improved safety equipment tend to provide a lower injury risk for occupants. This is not seen in the current study, based on currently defined assessment methods. With regard to employing the current assessment criteria as defined in EN 1317 or MASH, testing with old test vehicles, and not taking into account improved vehicle safety equipment (such as seat belt pretensioners, load limiters, or even a simple airbags), does not result in a disadvantage to the assessment of impact severity. The question arises in this context of whether the criteria used are truly providing an accurate assessment of the occupant injury risk: are any

other criteria, such as HIC (Head Injury Criterion), a3ms, chest acceleration, etc.—as used in vehicle safety—that would be better for correctly assessing the occupant risk?

**Author Contributions:** Conceptualization: E.T.; Methodology: E.T. and G.G.; Formal analysis: E.T. and G.G.; Writing—original draft preparation: E.T.; Writing—review and editing: G.G.; Project administration: E.T.; Funding acquisition: E.T. All authors have read and agreed to the published version of the manuscript.

**Funding:** This research was funded by the Austrian Research Promotion Agency (FFG) "Mobilität der Zukunft, Ausschreibung Verkehrsinfrastrukturforschung 2018" tender grant number 873194.

**Institutional Review Board Statement:** Not applicable.

**Informed Consent Statement:** Not applicable.

**Data Availability Statement:** No report on further data.

**Acknowledgments:** Open Access Funding by the Graz University of Technology.

**Conflicts of Interest:** The authors declare no conflicts of interest.

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
