# Peer review of "Impacts on Crash Cushions—Analysis of the Safety Performance of Passenger Cars with Improved Safety Equipment Compared with Test Vehicles Based on Assessment Criteria as Defined in EN 1317"

_infrastructures, doi:10.3390/infrastructures9030059_

Round 1

Reviewer 1 Report

Comments and Suggestions for Authors

good article, interesting to read

some comments for improvement inside pdf

Author Response

Thank you very much for your valuable comments on the manuscript. We were able to significantly improve the quality as a result.

abrevasions to be explained at first use

added in the abstract

please define a clear research question at introduction where you can state the clear answer to this question in conclusion

the objective of the study was given at the end of the literature section. We have enhanced the text and added the assessment criteria.

don't split figure and figure description

Unfortunately not referenced in the text and picture not positioned appropriate. Moved to the description in the text and referenced.

figure formating not acceptable,

figure width bigger than text width

the figure positioned follows the template scheme given by MDPI.

gap between back-up and wall! (not cushion)

changed

figure width to wide and figure underline splitted from figure

the figure positioned follows the template scheme. Picture and figure caption is not splitted.

graph width

not quite sure what is meant but the pictures follows the scheme of the template

source of this equation? is it [1]?

explain the terms used inside the equation

the source is [1] and the terms are explained in the text previous to the equation.

same as equation 1

explained in the text

same as equation 1 and 2

explained in the text

Reviewer 2 Report

Comments and Suggestions for Authors

MDPI – Infrastructures

Impacts on crash cushions – Analysis of the safety performance of passenger cars with improved safety equipment compared with test vehicles based on assessment criteria as defined in EN 1317

Reviewer #1

The paper aims to assess the occupant safety of passenger cars in terms of examining impact configurations against a crash cushion that deviates from the EN 1317 approval tests.

The methodology is sound and the conclusions are adequately supported by the experiment results. Is there any additional testing that should be performed in order to verify the authors’ standpoints?

Based on the performed tests, the authors conclude with certain questions for further research. What should these tests comprise of in terms of prioritization?

Please find below certain additional comments.

The Figures initially should be introduced in the text and then displayed with a caption (e.g. Figure 2).

The impact design speed for the crash cushion was 80km/h. On what basis was the 100km/h impact speed selected?

Author Response

Thank you very much for your valuable comments on the manuscript. We were able to significantly improve the quality as a result.

The methodology is sound and the conclusions are adequately supported by the experiment results. Is there any additional testing that should be performed in order to verify the authors’ standpoints?

We are not fully understand the reviewer comments. Additional testing would be encouraged. We are lacking of tests on road restraint systems according to EN1317-2 with an oblique test and different passenger cars, i.e. improved occupant safety equipment.

Based on the performed tests, the authors conclude with certain questions for further research. What should these tests comprise of in terms of prioritization?

We did not want to propose a test or to priories a test. We have tried to foster discussion about the assessment criteria in EN 1317. However, the first part of the study results should only report the standard criteria of EN 1317. In a subsequent paper we would analyse the vehicle criteria and compare that with criteria of EN 1317. This paper is under preparation and will be submitted soon.

Please find below certain additional comments.

The Figures initially should be introduced in the text and then displayed with a caption (e.g. Figure 2).

Unfortunately not referenced in the text. Reference is now given and the picture positioned appropriate. All other pictures and tables checked and referenced if necessary.

The impact design speed for the crash cushion was 80km/h. On what basis was the 100km/h impact speed selected?

The selected speed was referenced to a previous study and is based on real-world accident analysis. We have modified the sentence to make it clearer.

Reviewer 3 Report

Comments and Suggestions for Authors

The article presented is interesting with respect to road safety assessment, with logistically difficult experiments. However, I have following comments: 

1. It would be suitable to declare also mass of the used crash cushion, not only its dimensions.

2. In the line 405 - it seems that authors used values for OIV when discussing THIV for offset impact.

3. When discussing THIV for offset impact beginning in the line 405 - authors state that higher value for VW Golf in comparison to Opel Corsa could be caused by oscillations in the acceleration signal and if these oscillation are smoothed then THIV value for VW Golf would be lower. Why should one further smooth acceleration signal if it was previously filtered?( if the tests were realized and THIV values were calculated according EN 1317). It is not explicitly stated by the authors but I assume frequency class CFC180 was used if EN 1317 requirements were followed.

4. I have reservations with respect to experimental setup - authors are comparing not only old and new ("safer") vehicles, but different mass categories when one vehicle is 25% heavier than another. The fact that the vehicle mass is not controlled quantity has significant influence on the results. Or should be vehicle mass also considered part of "improved safety equipment"?

5. The goal of the article - showing of inadequacy of EN 1317 or MASH safety criteria for crash cushion safety assessment - would be better served if these criteria would be compared with the dummy based criteria, particularly when the vehicles were equipped with Hybrid III ATD. Was the ATD instrumented?

Comments on the Quality of English Language

The English language is overall fine.

Author Response

Thank you very much for your valuable comments on the manuscript. We were able to significantly improve the quality as a result.

The article presented is interesting with respect to road safety assessment, with logistically difficult experiments. However, I have following comments: 

  1. It would be suitable to declare also mass of the used crash cushion, not only its dimensions.

We have included weight in the experimental set-up description.

  1. In the line 405 - it seems that authors used values for OIV when discussing THIV for offset impact.

Thank you for very precisely reviewing the text. Indeed we used the wrong values.

  1. When discussing THIV for offset impact beginning in the line 405 - authors state that higher value for VW Golf in comparison to Opel Corsa could be caused by oscillations in the acceleration signal and if these oscillation are smoothed then THIV value for VW Golf would be lower. Why should one further smooth acceleration signal if it was previously filtered?( if the tests were realized and THIV values were calculated according EN 1317). It is not explicitly stated by the authors but I assume frequency class CFC180 was used if EN 1317 requirements were followed.

Indeed, it is difficult to explain that without the signal. We included a picture of THIV vs. displacement to make it more clear. The analysis was done in accordance with EN1317.

  1. I have reservations with respect to experimental setup - authors are comparing not only old and new ("safer") vehicles, but different mass categories when one vehicle is 25% heavier than another. The fact that the vehicle mass is not controlled quantity has significant influence on the results. Or should be vehicle mass also considered part of "improved safety equipment"?

This is a very good argument. The newer vehicle has much more weight 1.245 kg compared to 935 kg. However, the criteria ASI and THIV and others did not show a huge difference, specifically in the full overlap configuration. For the offset there were differences between the vehicles, positive (lower values) for ASI, PHD and ORA in the case of the heavier vehicle but negative (higher values) for THIV and OIV. We have discussed this but added some additional text in the discussion section. The weight should have an impact on safety, but we do not believe that this is “improved safety equipment” due to the fact that the main assessment criteria in EN 1317 – ASI and THIV in the full overlap did not show a difference between the two vehicles. In the offset test the ASI of the vehicle with lower weight was 1.25 times higher (2.6 compared to 2.1) and the mass ratio is 1.33 (1245 compared to 935). this is somewhat in a similar range. The THIV, however would be 0.85 times (43.3 compared to 51.6) for the vehicle with lower weight compared to the heavier.

  1. The goal of the article - showing of inadequacy of EN 1317 or MASH safety criteria for crash cushion safety assessment - would be better served if these criteria would be compared with the dummy based criteria, particularly when the vehicles were equipped with Hybrid III ATD. Was the ATD instrumented?

Yes, we did this. However, the first part of the study results should only report the standard criteria of EN 1317. In a subsequent paper we would analyse the vehicle criteria and compare that with criteria of EN 1317. This paper is under preparation and will be submitted soon.

Round 2

Reviewer 3 Report

Comments and Suggestions for Authors

I consider points 1,2 and 5 of my original review answered. With respect to the points 3 and 4:

Ad 3: Authors presented curves for THIV-displacement in the figure 12b - it seems that the y values are in m/s and not in km/h. However, my previous question still stands unexplained - why would be additional filtering/smoothing of the acceleration signal be necessary if the signal was already filtered using CFC180 according EN1317? Is this approach methodically sound? If the authors can not explain this, then they should limit their speculation in this part of the article wrt. of differences between THIV values in full and partial overlap, particularly when only limited set of tests was performed. Rephrasing of the relevant sentences of the article is advised.

Ad 4: The weight of vehicles clearly has an influence - but it is "compensated" by the different stiffness of vehicles. In full overlap - the values for both vehicles are similar because although worse condition for Opel momentum change is to be expected due its lower weight, it is "softer", absorbs more deformation (wheelbase shortening, header rail buckling) and decelerates during longer time interval. If the weight of the vehicles was controlled and "old" vehicle with similar weight as VW was used, then paradoxically lower values of AIS, etc. should be expected for older vehicle. This should be discussed in the article, at least for full overlap tests. In partial overlap it seems that the complex interactions between stiffness and weight of vehicle caused that no hard conclusion wrt. the values could be made - again probably caused by not controlling the weight (or vehicle stiffness if influence of weight on values was to be researched).  However I agreed with the authors that currently defined criteria in EN1317 and MASH are not sufficient for accurate evaluation of crash cushions, this can be seen from the performed tests if properly analyzed.

The article should have mentioned limitations of the study, where not sufficient control of the vehicle weight and low number of tests should be mentioned.

Comments on the Quality of English Language

English language is overall fine.

Author Response

Dear reviewer,

thank you for the detailled response. I have explained in the document and included changes, rephrasing in the manuscript.
